# Experimental and Numerical Investigations of Titanium Deposition for Cold Spray Additive Manufacturing as a Function of Standoff Distance

**DOI:** 10.3390/ma14195492

**Published:** 2021-09-23

**Authors:** Wojciech Żórawski, Rafał Molak, Janusz Mądry, Jarosław Sienicki, Anna Góral, Medard Makrenek, Mieczysław Scendo, Romuald Dobosz

**Affiliations:** 1Faculty of Mechatronics and Mechanical Engineering, Kielce University of Technology, Tysiąclecia Państwa Polskiego 7, 25-314 Kielce, Poland; 2Faculty of Mechanical Engineering, Bialystok University of Technology, Wiejska 45c, 15-351 Białystok, Poland; r.molak@pb.edu.pl; 3Faculty of Materials Science and Engineering, Warsaw University of Technology, Wołoska 141, 02-507 Warszawa, Poland; romuald.dobosz@pw.edu.pl; 4Polskie Zakłady Lotnicze Sp. z o.o., Wojska Polskiego 3, 39-300 Mielec, Poland; janusz.madry@lmco.com (J.M.); jaroslaw.sienicki@lmco.com (J.S.); 5Institute of Metallurgy and Materials Science, Polish Academy of Sciences, Reymonta 25, 30-059 Kraków, Poland; a.goral@imim.pl; 6Faculty of Management and Computer Modeling, Kielce University of Technology, Tysiąclecia Państwa Polskiego 7, 25-314 Kielce, Poland; fizmm@tu.kielce.pl; 7Institute of Chemistry, Jan Kochanowski University of Kielce, Żeromskiego 5, 25-406 Kielce, Poland; scendo@ujk.edu.pl

**Keywords:** cold spraying, titanium, additive manufacturing, standoff distance, deposition efficiency

## Abstract

In this research, the cold spray process as an additive manufacturing method was applied to deposit thick titanium coatings onto 7075 aluminium alloy. An analysis of changes in the microstructure and mechanical properties of the coatings depending on the standoff distance was carried out to obtain the maximum deposition efficiency. The process parameters were selected in such a way as to ensure the spraying of irregular titanium powder at the highest velocity and temperature and changing the standoff distance from 20 to 100 mm. Experimental studies demonstrated that the standoff distance had a significant effect on the microstructure of the coatings and their adhesion. Moreover, its rise significantly increased the deposition efficiency. The standoff distance also significantly affected the coating microstructure and their adhesion to the substrate, but did not cause any changes in their phase composition. The standoff distance also influenced the coating porosity, which first decreased to a minimum level of 0.2% and then increased significantly to 9.8%. At the same time, the hardness of the coatings increased by 30%. Numerical simulations confirmed the results of the tests.

## 1. Introduction

New low-cost manufacturing technologies are the basis of further development for many branches of industry. Among them, additive manufacturing processes play a unique role as they allow parts of machines to be manufactured based on layer-by-layer deposition. Combined with 3D modelling and automation, these processes allow components to be produced with complex shapes and, additionally, significantly lower time and costs [1,2,3]. 

In recent years, cold spraying has joined the group of applied additive technologies, such as selective laser sintering (SLS) or direct metal deposition (DMD) [4,5,6,7]. “In this process compressed and heated gas accelerate particles to supersonic speed in the de Laval nozzle. As the gas flows through the divergent part of the nozzle, it decomposes to a significant temperature drop, even below the ambient temperature, so the particles remain solid throughout the process. The detailed basics of the cold spray process are widely presented in literature [8,9,10]. The low process temperature does not cause any phase changes in the deposited coating which is especially important when oxygen-sensitive materials are used.” Such materials include titanium, which has a high-oxygen affinity that creates problems with processing, necessitating the use of expensive production processes in a controlled atmosphere, such as vacuum melting [11]. On the other hand, titanium with a high strength to weight ratio and excellent corrosion resistance in many media, including seawater, is an irreplaceable material for many applications in aerospace [12,13,14]. 

The technologies currently used for manufacturing titanium components, which involve casting, forging, extrusion, and machining, are expensive and labour-consuming. Moreover, the production processes of many parts lead to significant losses of material, which can reach up to 60%. Therefore, direct producing of titanium is crucial especially in the aviation industry [15,16,17]. 

The properties of cold-sprayed coatings are particularly sensitive to the spraying parameters because the coating is formed from solid particles. The problem is that impacts during the cold spray process occur in a very short time and real-time observations are virtually impossible. Numerical modelling supports the selection of parameters and allows us to understand the deposition process. Particle vs. substrate interactions during the CS and resultant bonding are important because they affect coating features. Many numerical simulations have been performed to study the impact and deformation process during CS under different variables such as material combination [18,19], impact velocity [19,20], particle diameter [21,22] or pre-heating temperature [23,24]. Numerical modelling confirmed that CS coating characteristics, i.e., the flattening ratio of the powder, deposition efficiency, and coating adhesion could be described as a function of the current ratio of the particle impact velocity to the critical velocity [25], which confirmed the need for numerical modelling of the CS process. The first model and the most widespread today is the Lagrangian simulation. This analysis is ideal for tracking material motion and deformation in regions of relatively small deformation. In addition, the framework mentioned above was also used to introduce the concept of adiabatic shear instability [26], and allowed us to understand phenomena occurring at the microstructural level. The following approach in modelling was the Eulerian framework, which was used to overcome the problem of highly distorted elements in the Lagrangian scheme [27]. Unfortunately, both models were not perfect and showed significant errors. Their limitations, mainly due to severely distorted elements, were overcome using the Arbitrary Lagrangian –Eulerian framework (ALE) with adaptive remeshing [28]. 

Typically, a maximum temperature and gas pressure are used to create the highest velocity of the powder grains. The problem is the spraying of metals with a low melting point, where excessively high speed can cause nozzle clogging [29]. However, in the case of metals with a high melting point, this problem does not occur, and the maximum pressure (50 bar) and maximum gas temperature (1100 °C) are used [30]. Therefore, the main factor determining the velocity and temperature of a particle at the moment of impact is the standoff distance, which directly affects its degree of deformation and thus the properties of the coating. From the numerical simulations conducted by Pattison et al. [31], it is clear that cold-sprayed coatings should be deposited at a standoff distance at which the gas velocity is higher than the velocity of the powder particles. When the spraying process is within a short distance of 10–20 mm, the presence of bow shock disrupts the formation of the coating, which in turn causes a decrease in deposition efficiency, while increasing the spraying distance causes the gas stream velocity to drop below the velocity of the sprayed particles. Then, a negative drag force arises that reduces their kinetic energy and thus the coating properties [31]. The standoff distance is an important parameter that directly affects the particle velocity at the moment of impact onto the surface and the deposition efficiency which is also an important parameter because it partially affects the cost of the deposit. Therefore, deposition efficiency has been the subject of many studies analysing various powders and parameters to obtain its maximum value. Gilmore et al. [32] studied the effects of Cu particle velocity and spray angle. The deposition efficiency is also influenced by the presence of oxides on the surface of Cu particles [33,34], and the presence of a thick oxide film on the surface of Inconel 625 cold-sprayed with CoNiCrAlY powder. The significant influence of particle size distribution was shown by Li et al. [35] with the example of Cu powder. An important role of powder composition in the spraying mixtures was presented for the 316 L stainless steel and Fe [36] and mixtures of B_4_C, TiC and WC carbides with Ni powder [37]. The importance of Cr powder hardness for deposition efficiency was demonstrated by Yeom et al. [38] through its annealing. The travel speed of the nozzle [39,40] and the composition of the He-N_2_ mixture were also analysed [41]. Several works were also devoted to modelling and numerical simulations of the deposition efficiency [39,42,43,44]. All the above experiments and analyses were carried out at an arbitrary assumed standoff distance. Literature on the influence of standoff distance is limited [31,45,46,47]. Moreover, there are no studies that concern the direct relationship between the standoff distance and deposition efficiency, which is particularly important in the case of titanium for cold spray additive manufacturing processes. Therefore, this research aimed to determine the influence of the standoff distance on the efficiency of the cold spraying process and numerical verification of the obtained results.

## 2. Experimental Details

### 2.1. Feedstock Powder and Substrate Material

Commercially pure titanium powder (99 wt.% Ti) was used as feedstock in this study. This powder was manufactured using the hydrite–dehydrite process and supplied by Kamb Import-Export (Warsaw, Poland The particle size distribution of this powder was tested with a HELOS H2398 laser diffractometer from Sympatec GmbH (Clausthal-Zellerfeld, Germany). The substrate was an aluminium alloy (Al 7075) with dimensions of 400 × 30 × 5 mm and the sample preparation before the cold spray process involved degreasing only.

### 2.2. Deposition of Coatings and Sample Preparation

The cold spray process was performed at Kielce University of Technology with an Impact Innovations 5/8 System (Impact-Innovations GmbH, Rattenkirchen, Germany) cooperating with a Fanuc M-20iA robot (Fanuc Robotics Ltd., Oshino, Japan). Nitrogen was applied as the process gas to deposit titanium coatings. In order to ensure the maximum velocity and temperature of the titanium powder grains, the maximum system parameters were used; temperature 800 °C and pressure 40 bar. The coatings were deposited with a variable stand of distance in the range of 20–100 mm, which was increased successively by 10 mm. The nozzle traverse speed was 400 mm/s and the deposition step size was 2 mm. In order to obtain the appropriate thickness of samples, twelve layers were deposited on each of the nine samples as an element of the additive manufacturing process. The deposition process involved cooling the specimens and keeping them at a temperature below 80 °C. 

### 2.3. Charcterisation of Microstructure and Phase Composition

A Jeol JSM-7100 SEM microscope (JEOL Ltd., Tokyo, Japan) was used to analyse the microstructure of the titanium powder and the deposited coatings. Prior to the cross-sectional analysis, the powder and coating samples were embedded in resin and then polished with increasingly fine, i.e., 3 µm, 1 µm and 0.25 µm, diamond suspensions. The morphology of the as-sprayed coatings was analysed using a Talysurf CCI-Lite non-contact 3D profiler (Taylor Hobson Ltd., Leicester, UK). The ImageJ program (ImageJ 1.48, NIH and LOCI, Bethesda, MD, USA) was applied to measure the porosity of the coatings, in accordance with ASTM-E2109. The phase composition of the as-sprayed deposits was studied using a Bruker D8 Discover diffractometer (Bruker Ltd., Malvern, UK), while that of the powder feedstock was investigated with a Philips X’Pert PW 1710 diffractometer (PANalytical, Almelo, The Netherlands) equipped with software. In both cases, Co-Kα radiation (λ = 1.78897 Å) was used. 

### 2.4. Evaluation of the Relative Deposition Efficiency

The measurement of the coating profile, which was the basis for the evaluation of the relative deposition efficiency, was carried out using a PG-2/200-3D mechanical profilometer (IOS, Kraków, Poland).

### 2.5. Characterisation of Mechanical Properties

The micromechanical tests were carried out on polished cross-sections of the deposited coatings using a Nanovea tester (Nanovea Inc, Irvine, CA, USA) with a Berkovich indenter (Olivier and Pharr methodology, Nanovea Inc, Irvine, CA, USA), 20 mN applied load and 50 mN/min loading rate. Forty-nine measurements were made on each cross-section of the cold-sprayed titanium coatings. The hardness (HV0.3) of the coatings was measured using an Innovatest Nexus 4000 tester (Innovatest, Maastricht, The Netherlands). Five readings were carried out for each coating. 

### 2.6. Numerical Simulations

The numerical analysis was divided into two independent parts. The first part concerned the behaviour of a single titanium particle with a diameter of 35 μm when hitting the surface of a flat plate made of 7075 aluminium alloy. The analysis was carried out using the Ansys Autodyn (ANSYS Inc., Canonsburg, PA, USA) environment, with the ALE approach (Arbitrary Lagrangian–Eulerian) and the SPH (Smooth Particle Hydrodynamics) method. The second part allowed for behaviour analysis of the gas stream and particles to be analysed after release of the de Laval nozzle to the moment of hitting the surface of the plate, which was the substrate for the titanium layer deposited. The calculations were made for a steady-state, assuming ideal gas conditions. The model includes the SST k-omega turbulence model and the energy equation. Particle movement was calculated using the DPM (Discrete Phase Model) model.

## 3. Results and Discussion 

### 3.1. Characterization of the Ti Powder

The titanium powder used in the experiment consists of irregular and angular grains as a result of its production by the hydrite-dehydrite process (Figure 1a). The large surface area per unit mass made the powder grains more susceptible to oxidation. The cross-sectional view in Figure 1b reveals that titanium grains are homogeneous and have no inclusions or internal porosity. On the basis of the analysis of the powder’s particle size distribution, it was found that it is characterised by the d10 = 18.00 μm, d50 = 35.00 μm and d90 = 60.00 μm parameters (Figure 2).

### 3.2. Microstructure of the Titanium Coatings 

The effect of the standoff distance on the microstructure of titanium coatings is shown in their cross-sections in Figure 3a–i (near substrate—upper picture, top—picture below). 

Significant changes in the microstructure of coatings together with increasing standoff distance in the range of 20 to 100 mm are visible. Porosity is the fundamental property of thermall-sprayed coatings, especially of cold-sprayed coatings, which are formed from deformed particles in a solid state. Porosity directly affects the coating microstructure as well as their mechanical and thermal properties [45,48]. Spraying at short distances (20 and 30 mm) produces porous coatings which adhere well to the aluminium alloy substrate. The reason for such increased porosity of the coatings in these cases may be the lower speed of larger particles, as indicated by the parameter d90 = 60 μm. Li et al. [48] reported a significant difference in velocity between fine and coarse particles of titanium angular powder after they leave the nozzle. After leaving the nozzle, the velocity of the powder grains increased depending on their size, reaching the highest value at a distance ranging from 50 to 120 mm. Initially, the fine powder grains have a speed considerably higher than that of large grains, which, due to their greater weight, accelerate more slowly. However, because of their small weight, their speed drops rapidly. A close standoff distance is sufficient for large particles to form good adhesion with the substrate, with no pores visible at the interface. However, it is not sufficient to create high cohesion between the deposited coatings and subsequent titanium particles, therefore their insufficient deformation causes the formation of pores. In addition, the bow shock phenomenon reduces the deposition of fine titanium grains that have too little kinetic energy to overcome it and reach the substrate to create a coating [31,49,50]. An increase in the standoff distance leads to a visible decrease in the porosity, which at 70 mm reaches a minimum value of 0.2% (Figure 3f). This coating is characterised by a small number of very fine pores that appear evenly over the entire cross-sectional area. At this distance, the maximum deformation of all powder grains, regardless of their size, creates the largest area of their connection. This is due the adiabatic shear phenomenon, resulting in high temperature and consequently local melting between the depositing particles of titanium powder [51]. Further increases in the standoff distance resulted in a significant increase in porosity, which at 100 mm reached the highest value of 9.8% (Figure 4). The velocity value of all the powder grains also decreased regardless of their dimensions, however, it was still above the critical velocity, which is a condition for the formation of the coating [52]. It can be assumed that the most significant decrease in velocity concerned fine powder grains, which deformed slightly, contributing to a significant increase in the porosity of the coating. 

On the other hand, the kinetic energy of the largest grains was large enough for an adiabatic shear process to occur upon impact with the previously deposited titanium layer, during which the high temperature enabled the formation of a metallurgical bond [53]. This porosity level is close to the value obtained for coatings sprayed from a distance of up to 40 mm [11,52,54]. The presence of pores is uniform over the entire area of all cross-sections of cold-sprayed titanium coatings, but fewer of them are visible in the upper part. Such changes in the porosity of the coatings sprayed with cold gas result from the course of the coating forming process, where the last layer is not subjected to penning as intensively as the layers previously deposited [45,52,55]. Visibly lower porosity in the last deposited layer occurs at all spraying distances from 20 to 100 mm. It can be assumed that all the grains of the titanium powder have a high speed, exceeding the critical speed, which allows them to be significantly deformed (Figure 3a–i, bottom). The coating porosity obtained during the experiments (0.2–9.8%) is similar to that reported for coatings sprayed using irregular titanium powder [11,13,51,53,54,56]. The conducted research leads to the conclusion that the most critical factor is the gas velocity, which depends on the type of gas, its temperature and pressure, and the second factor is the particle size distribution. Nevertheless, considerable variations in porosity are reported using the same spray parameters. Zahiri et al. [11] indicated that cold-sprayed fine particles of titanium (16 μm) deposited in the presence of helium at a standoff distance of 20 mm formed coatings with a porosity reduced to 0.5%. Moy et al. [54] reported porosity in the range of 8.9–12.1% for coatings sprayed using powder with an average particle size of 22 μm. The porosity of cold-sprayed Ti coatings exceeding 20% is reported by Marrocco et al. [51]. Furthermore, it was shown that porosity was independent of the gas pressure. The findings are not consistent with the results presented by Gulizia et al. [57]. 

Figure 3a–i (top) show the interface; cold-sprayed titanium coating–AA7075 substrate. The influence of the standoff distance on the adhesion of the coatings is clearly visible. Coatings deposited at close distances (20–40 mm) adhere very well to the substrate, with only trace discontinuities detected. On the other hand, spraying at the upper distance range (80–100 mm) leads to the formation of clearly visible porosity at the interface, which reduces adhesion. The shape of the coating–AA7075 substrate interface observed in the upper pictures in Figure 3a–i was similar in all cases. It can be assumed that the kinetic energy of particles was sufficiently high to crater the substrate and build up the first layer, but the degree of particle deformation and realignment was lower than that reported at shorter distances [58]. 

To show the microstructure features, cross-sections of some coatings sprayed at the standoff distance of 20 mm, 70 mm, and 100 mm were etched (Figure 5). Microstructures of the coating after polishing are fraught with error because the actual image undergoes partial distortion and etching reveals more details of coatings. Significant differences in coating porosity depending on the standoff distance can be clearly seen. Images obtained after etching corroborate with corresponding microstructures of coatings after polishing. The presence of pores with different shapes in coatings sprayed with the shortest and longest standoff distance can be attributed to the angular morphology of the sprayed powder and lower speed of grains insufficient to fill in the unevenness of the surface accurately. On the other hand, only slight porosity in the form of very small points can be found in the coating sprayed at the distance of 70 mm (Figure 5b). Moreover, taking into account the irregular and angular shape of the feedstock, differences in the degree of deformation of powder grains can be observed. Very poorly deformed large grains are visible in the microstructures of coatings deposited with both extreme distances (Figure 5a,c—white arrows). Many poorly deformed grains are present in the coating sprayed at a standoff distance of 100 mm (Figure 5c). Lower speed reduces their kinetic energy and ability to plastic deformation when they impact with the substrate. Consequently, it is clearly visible that interfaces between the deposited particles depend on the standoff distance.

It is worth noting that, despite the wide size distribution of the powder (below 18 and over 60 mm), the microstructure of the titanium coating sprayed at a standoff distance of 70 mm (Figure 5b) consists of the most severely plastically deformed grains. Tight bonds are clearly seen between highly compacted irregular particles of different sizes. Interparticle bonding features are clearly visible at higher magnification boundaries among grains (white arrows), showing that coatings consist of heavy deformed titanium particles (Figure 5d). The presented dense microstructure may indicate metallurgical bonding between grains due to a high interfacial temperature that arises during the phenomenon of adiabatic shear instability [9]. Such intimate amalgamations are only partially present in the other two cases. Areas where tight bonds occur are clear, however, areas at random locations where partial bonding among particles is well seen (Figure 5a,c—black arrows). Such imperfections in coating microstructure decrease the coating cohesion.

Surface examinations revealed that the cold-sprayed titanium coatings obtained from irregular powder were characterised by high roughness (Figure 6a–i). Despite the high velocity of striking particles, the surface was very rough because of the use of powder with irregular particles (Figure 3a–i—bottom). There were no Ti splats, like in the case reported by Moy et al. [54]. The changes of the porosity inside the cold-sprayed titanium coatings dependent on the standoff distance were consistent with changes in the porosity observed at the surface. Large pores were reported both at short and long distances (Figure 6a,i, respectively). However, the coating with the lowest porosity (Figure 6f) showed negligible porosity at the surface too.

### 3.3. Influence of the Standoff Distance on Coating Hardness and Elastic Modulus

Figure 7 shows how the standoff distance affects the microhardness, nanohardness, and elastic modulus of the Ti coatings. The effect of porosity on the mechanical properties of the cold-sprayed coatings is also visible. As depicted in Figure 4, an increase in the standoff distance resulted in a decrease in porosity and an improvement in the mechanical properties of the material; their highest values were observed at a standoff distance of 70 mm. A further increase in the standoff distance, however, led to higher porosity and worse mechanical properties. Similar observations were reported by [17,49,59]. Nevertheless, there are researchers who have not confirmed this relationship [60,61].

Most measurement results show a high standard deviation because the microstructure of cold-sprayed coatings consists of particles differing in their degree of deformation. This is due to a wide particle size distribution and different particle positions in the jet, leading to different indentation values dependent on the particle size. Moreover, poorly bonded particles make microhardness measurement difficult [56]. At high porosity, microhardness is more a function of porosity than of hardness. It is thus more reliable to measure nanohardness in different regions [62]. The nanohardness and elastic modulus maps for Ti coatings deposited at a standoff distance of 20 mm are presented in Figure 8. The maps show that there are substantial differences in the mechanical properties of cold-sprayed titanium coatings in microareas. Local variations in the mechanical properties are caused by severe plastic deformation of particles at the moment of impact onto a surface, and consequently, a local increase in hardness [63] and pores under the surface. The shot peening effect (particles rebounding off the surface) additionally strengthens the coating microstructure, increasing its hardness [45,48,51]. However, the results obtained from 49 measurements confirmed the above-mentioned relationships concerning differences in the obtained results of microhardness.

### 3.4. Phase Composition of Cold-Sprayed Ti Coatings

Figure 9 presents the X-ray diffraction patterns recorded for the titanium powder and cold-sprayed titanium coatings. The XRD results indicate that there are no new phases in the coating when compared with the starting feedstock. Analysis of the diffraction patterns obtained for the Ti powder and all the cold-sprayed coatings reveals that only the crystalline phase is present in both cases. It can be seen that significant increases in the standoff distance (up to 100 mm) and high gas temperature (800 °C) do not cause the formation of oxides, even though powder with irregular particles has a greater surface area per unit mass than powder with spheroidal particles. The SEM EDX analysis carried out at five points for each of the cold-sprayed titanium coatings revealed that there was no oxygen present. 

Figure 10a shows a SEM image of a Ti coating deposited at a standoff distance of 100 mm, while Figure 10b provides details of the EDX analysis for point 1 (white cross).

The initially high temperature of the powder in the convergent part of the nozzle drops rapidly after particles enter the divergent section of the nozzle and then, again, after they leave it [64]. The contact time of the powder particles with the hot process gas is very short and the rate of decrease in the gas temperature in the divergent part of the nozzle is much higher. Raoelison [65] reported that the time for the particles to pass through the nozzle was about 10^−7^–10^−6^ s and the total time to hit the surface did not exceed 10^−3^ s; however, the temperature of particles depended on their size (the bigger the particle, the higher the temperature) [64]. Since, throughout the process, the particle temperature was much lower than the very high initial gas temperature, it was also much lower than the melting point of the feedstock material. The coating was thus formed from particles in a solid state. Therefore, the main advantage of cold-sprayed coatings is the lack of defects typical of thermally sprayed coatings, such as oxidation, evaporation, gas release, and shrinkage porosity. The absence of oxides in cold-sprayed titanium coatings has also been reported by other researchers in the case of powders with angular morphology but different particle size distribution applied as the feedstock material [54,56,66,67,68,69]. However, some reports show a negligible increase in the contents of oxygen and nitrogen in the cold-sprayed coatings in comparison with the starting feedstock [52,57]. 

### 3.5. Surface Topography of the Coatings

The surface topography of the Ti coatings was characterised by height parameters providing information about their surface roughness. These parameters were calculated using all the measurement data from the optically scanned surface area, according to ISO 25178 (Table 1). 

The parameters were derived from the Abbot–Firestone curve, which characterises the functional behaviour of the surface measured. The height variations described by the arithmetic mean height Sa are the highest (23.40 µm) for the coating sprayed at the standoff distance of 30 mm and the lowest (16.20 µm) at 80 mm. As can be seen from Table 1, the differences in the values of Sa for the examined Ti coatings were distinct but not dependent on the distance between the spray gun and the sprayed substrate. No clear relationship between Sa and this process parameter was observed. The case is similar for the root mean square deviation (Sq). The two key parameters used to characterise the asymmetry and flatness of the surface are the skewness (Ssk) and kurtosis (Sku). In the case of the Ti coatings obtained at 20 mm, 30 mm, 40 mm, 50 mm and 80 mm, the surfaces exhibited asymmetry with a positive skewness of surface heights; the value obtained for the deposit at 80 mm (0.34) is up to three times higher than the values reported for the other coatings. The positive asymmetric roughness indicates that many hills and peaks are protruding from a mostly planar surface and the highest ones are found at 80 mm. On the other hand, the deposit obtained at 70 mm reveals a negative asymmetric roughness (−0.31) because the examined surface contains many pits. The other examined coatings have a surface with random roughness and Ssk is close to zero. For all the samples considered, the values of the parameter Sku are close to 3, but they are higher for those sprayed at 70 mm, 80 mm and 90 mm, and lower for the others. This indicates that the surfaces of these coatings gradually varied and free from extreme peaks or valley features. From the comparison of the Sp, Sv, and Sz parameters calculated from the absolute highest and lowest points, it can be concluded that the highest (Sp) and shallowest (Sv) points are found on the surface of the Ti coatings deposited at standoff distances of 20 mm and 50 mm. The other deposits reveal higher valley depths and lower peak heights.

### 3.6. Deposition Efficiency

Figure 11a shows the relationship between the cross-sectional area of the Ti coatings deposited by cold spraying and a standoff distance in the range of 20–100 mm.

It is evident that the influence of the standoff distance on the deposition efficiency is significant. The relative deposition efficiency (determined in relation to the cross-sectional area of the coating sprayed at a standoff distance of 20 mm) increased significantly at 50 mm and then rose steadily to reach a maximum at 100 mm. At 100 mm, the relative deposition efficiency was 53% greater than that reported at 20 mm. The thickness and width of the titanium coatings increased to 55% and 17% at 100 mm in comparison to the standoff distance of 20 mm (Figure 11b). The increase in the efficiency of the cold spray deposition with increasing standoff distance (from 20 to 70 mm) can be attributed to two phenomena. One is bow shock, described in Section 3.2. It is reported to significantly hinder the deposition of fine particles because their kinetic energy is too low to overcome compressed bow shock. Decelerated fine particles are not deposited onto the substrate surface [31,64]. Fukumoto et al. [49] indicated the significant influence of bow shock on the deposition efficiency; they found that it was much higher (eight times higher) when a special nozzle was used. An increase in the standoff distance reduced the influence of bow shock, causing an increase in the number of fine particles deposited and an improvement in the deposition efficiency. The other phenomenon is the increase in the particle velocity with increasing standoff distance [45]. It can be assumed that most particles significantly exceed the critical velocity at a standoff distance of 50 mm. A further increase in the particle velocity results in minimum porosity (0.5%). Fine titanium particles are easy to deposit; thermal diffusion does not limit their bonding, as is the case of tin, copper, or gold particles. Moreover, the range of critical velocity for titanium is very wide (690–890 m/s) [70]. It should be noted that, when the standoff distance was in the range of 20–70 mm, the amount of titanium powder deposited was higher because the coating porosity decreased substantially from 6.3% to 0.2%. The further slight increase in the deposition efficiency at a distance of 100 mm was associated with a considerable increase in the porosity of the cold-sprayed coatings. When the particle velocity exceeds a critical value, it is possible to create metallurgically bonded areas [25]. A similar porous structure was described by [45,52]. As the content of oxygen in the sprayed coatings was found to be higher, the authors suggested that, during spraying, the particles reacted with oxygen (flashing jet) and, during deposition, bonded together undergoing slight contact deformation caused by the adiabatic shear impacting phenomenon and, in consequence, limiting the contact area. Metallurgical bonding is the reason for the high efficiency of deposition of titanium particles even though their velocity was not high [52]. It can be concluded that, at a standoff distance greater than 100 mm, the gas velocity is lower than the particle velocity and the negative drag force causes the powder particles to decelerate. 

On the other hand, the actual number of the sprayed titanium powder particles seems to be smaller because the porosity grows markedly up to 9.8%. Studies on this subject provide divergent results. Gilmore et al. [32] suggest that the in-flight particle characteristics for the cold spray process change very little with varying standoff distance. Moreover, Lima et al. [71] reported that, when the standoff distance is shortened from 20 mm to 5 mm and the gas temperature increases, the deposition efficiency increases too. The results presented in this article are not consistent with the data provided by Li et al. [45]. They reported a considerable decrease in the deposition efficiency for titanium with increasing standoff distance, despite an increase in the particle velocity. The divergent results could be due to the use of much lower gas temperatures and pressures in the cold spray process. In the cold spray process, the deposition efficiency is directly dependent on the gas temperature and gas pressure, because an increase in the gas velocity causes the particles to accelerate. Most of the particles do not exceed the critical velocity (690 m/s), which results in lower deposition efficiency of titanium and high porosity of the coatings. Moreover, the application of finer titanium powder (d50 = 22.4 μm) with particles able to reach higher velocities than coarser particles did not cause a decrease in the deposition efficiency [45]. 

Assuming that the typical standoff distance is in the range of 10–50 mm [51], the experimental results obtained in this study for a distance of up to 100 mm (twice the maximum standoff distance) do not confirm the conclusion drawn by Rokni et al. [63]; that long standoff distances lower the deposition efficiency. However, is it essential that the thermal behaviour of particles during spraying are taken into account. Small particles are heated by the working gas very quickly because their thermal capacity is very low. Their cooling also occurs rapidly because there is a sudden drop in the gas temperature in the divergent part of the nozzle [72]. When the temperature of small particles is lower, the flattening of particles is insufficient, which results in higher porosity and worse mechanical properties. Particles with large dimensions heat up much more slowly because of a larger volume and can accumulate more heat over a longer distance. They lose little thermal energy, so their temperature decreases very slowly on the way to the surface. Moreover, the high velocity of large particles implies a shorter period of their presence in the gas stream, which reduces losses of heat. When bigger particles with thermal energy higher than that of smaller particles hit the surface, their temperature is higher. Combined with a high velocity, this leads to an increase in the deposition efficiency. Each case of cold spraying should be considered separately, taking into account the spraying system, the powder properties, and the process parameters, e.g., standoff distance. 

Figure 12 shows how an increase in the standoff distance in the range of 20–100 mm affected the microstructure of the titanium cold-sprayed coatings. As can be seen, there are three different microstructures dependent on the standoff distance. The first microstructure, resulting from a short distance deposition, contains only coarse deformed particles; fine particles were decelerated by bow shock, which increased the coating porosity. The second microstructure was obtained at a medium spraying distance. As most of the powder particles were deposited, the coating porosity was negligible, and the deposition efficiency increased. The final microstructure was produced at a long spraying distance. There were much fewer deformed particles, which caused the coating porosity and the deposition efficiency to increase (Figure 11a). It is difficult to precisely determine how the deposition efficiency and the properties of cold-sprayed coatings are affected by the velocity and temperature of irregularly shaped particles depending on the standoff distance.

## 4. Numerical Simulations

In order to take a closer look at the deposition process of particles on the substrate, two types of simulations were carried out. The first concerned the contact of a single titanium particle of spherical shape with the substrate of Al 7075. The result of these calculations is the analysis of the behaviour of the particle–substrate system at different velocities of the particle at the moment of its impact, which was the basis of estimating the optimal impact velocity. Due to the occurrence of large deformations and the short time of the analysed process, the simulations were carried out on the basis of the SPH meshless method and ALE (Arbitrary Lagrangian–Eulerian) approach in the Ansys AUTODYN software. The second type of simulation concerned the analysis of the distribution of the speed at which particles hit the substrate, taking into account the different distances of the nozzle. Ansys Fluent was used to perform this calculation.

### 4.1. Particle–Substrate Contact Simulations 

The analysis involves the impact of a single spherical particle with a diameter of 35 μm (d50 = 35.00 μm) on the substrate at an initial velocity of 400, 500, …, 900 m/s. Due to the symmetry of geometry and loads, axis-symmetric models were used. The particle was modelled by SPH approach while the finite element method in the ALE formulation was used to describe the behaviour of the substrate. The results of the simulation show deformations of the particle–substrate system and maps of equivalent stresses according to the Huber–Mises–Hencky theory.

Figure 13 summarizes the results of the first part of the performed numerical analysis in the form of cross-sections of a titanium powder particle in interaction with the surface of the aluminium alloy plate after 100 ns from the moment of contact with its surface. The analysis was carried out as a function of velocity of the particle (Vp) at the moment of collision, which varied in the range of 400 to 900 m/s.

Critical velocity is a physical quantity depending on both the physical and mechanical properties of the particle and substrate materials, particle geometry and technological parameters of the cold spray process (temperature, pressure, gas type, nozzle geometry, standoff distance), which has been widely discussed in many papers [25,73,74]. Particles sped up below the critical velocity value will be reflected from the substrate [8,75]. The reflection phenomenon is attributed to the elastic deformation energy stored in the substrate material which counteracts adhesion forces. Particles accelerated to velocities exceeding the critical value will be removed from the ground due to erosion. It is assumed that this value is, on average, about 2V_crit_. Titanium velocities of about 700–900 m/s [70] were indicated as the critical velocity, but it should be remembered that this is the value determined for certain fixed conditions of the cold spray process. As we can see after 100 ns from the moment of the incident, for particles with a velocity below 600 m/s (Figure 13a,b), a space appears in the plane perpendicular to the direction of impact. This shows limited adhesion and the tendency to reflect the particle from the base as a result of the elastic energy of the substrate impacts exceeding the adhesion value. The optimal conditions were observed in the velocity range from 600 to 700 m/s (Figure 13c,d). This is the window of deposition for which the requirements for adhesion occur and it should result in the coating with the lowest porosity. Further increasing of the particle velocity results in the appearance of severe shear plastic deformations in the form of an interfacial jet, resulting in the detachment of particle fragments from the substrate, which may contribute to the conditions necessary to produce a dense and discontinuous coating being disturbed (Figure 13e). The second effect in the velocity range above 700 m/s is space (porosity) reappearance in the perpendicular plane. As in the case of low velocity, the elastic energy of the substrate is mainly responsible for the observed phenomena. This effect may contribute to the re-growth of coating porosity because of leaving the window of deposition. 

### 4.2. Simulation of Particles Acceleration 

The estimated optimal impact velocity of the particle was used to estimate the optimal standoff distance. The Computational Fluid Dynamics simulations contained the gas stream and particles after release of the de Laval nozzle to the moment of hitting the surface of the plate. The calculations were made for a steady-state, assuming ideal gas conditions with the SST k-omega turbulence model and the energy equation. The axisymmetric fluid domain contains a nozzle with an outlet diameter of 13 mm and a space between the nozzle and the substrate: 100 mm diameter and an axial dimension of 20, 40, …, 100 mm. The grid resolution guaranteed y+ < 5. Simulation parameters at the inlet were consistent with the experiments: fluid -N_2_, pressure 40 bar, temperature 800 °C. For pressure–velocity coupling coupled scheme was used and Second Order Upwind as the discretization schemes. Solutions also used the pseudo transient under-relaxation method. Particle movement was calculated using the DPM (Discrete Phase Model) model with surface injection type 15.6 g/min of titanium particles with constant diameter, 35 μm. The non-spherical drag law was used where the shape factor equals 0.3, based on geometry of particles and initial calculations. The boundary conditions for particles were set as escape, because the main result of these simulations is impact velocity of particles not detailed simulations of deposition process.

Figure 14 shows the velocity profiles (resultant of axial and radial velocities) of the gas stream (left side) and deposited particles (right side) as a function of the standoff distance. Low distances from the substrate, i.e., 20 and 40 mm (Figure 14A,B), resulted in a very high gas and particle velocity, well above the window of deposition determined in the first part of the numerical analysis. 

As a result, a high porosity coating and low deposition efficiency were obtained at these cold spray process conditions. Situations at such a short standoff distance are also complicated by the phenomenon of bow shock, which may also affect the deposition efficiency of titanium particles. The experiment carried out in [31] showed the possibility of an apparent decrease in deposition efficiency due to the mentioned effect by up to 40% when the standoff distance is reduced below 60 mm. From a standoff distance of about 60–70 mm (Figure 14C), the particle velocity range seems to be the most favourable to achieve the window of deposition resulting in the lowest coating porosity while maintaining a relatively high deposition efficiency. Further increases in standoff distance (80–100 mm) reduce the kinetic energy of the particles because of the significant expansion of the gas stream and the consequent decrease in their tendency to deform, which contributes to a slight increase in porosity (Figure 14D). However, their energy will be large enough to overcome the impact of the substrate elastic energy and ensure adequate adhesion.

The performed numerical simulations confirmed the results obtained in the experimental part of the manuscript. The lowest porosity of titanium coatings was achieved for a distance of 70 mm (Figure 4) and the lowest deposition efficiency was observed for a short standoff distance, accompanied by a very high particle velocity and bow shock phenomenon.

## 5. Conclusions 

The work carried out presents experimental and numerical investigations of titanium cold-sprayed deposition with different standoff distances. The microstructure and mechanical properties of deposits on 7075 aluminium alloy concerning the deposition efficiency for use in the cold spray additive manufacturing process have been examined. The conducted research allowed the formulation of the following conclusions:The standoff distance in the cold spray process has a significant influence on the porosity of the coating. At the closest distance of 20 mm, the porosity was 5%. Then, at the distance of 70 mm, it dropped to the lowest level of 0.2%. Increasing the standoff distance to 100 mm caused its most significant increase to 9.8%.The mechanical properties of the deposit (microhardness, nanohardness, modulus of elasticity) also depend significantly on the standoff distance. The highest level of their value was achieved for deposits obtained with a standoff distance of 70 mm.There were no phase changes in the phase composition of the titanium deposits due to the increased standoff distance.The surface topography of the cold-sprayed titanium coatings shows significant differences, but no clear influence of the standoff distance is observed.The relative deposition efficiency increases with increasing standoff distance; at 100 mm it was 53% greater than at 20 mm.The obtained results of the experiments were confirmed by the numerical simulation of the cold spray process at different standoff distances.

## Figures and Tables

**Figure 1 materials-14-05492-f001:**
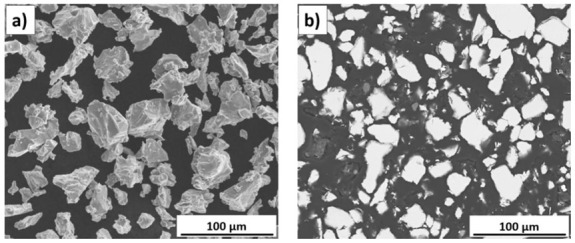
The titanium powder: (**a**) morphology, (**b**) cross-section.

**Figure 2 materials-14-05492-f002:**
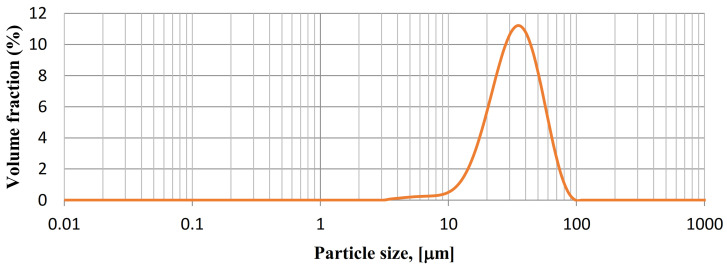
Particle size distribution of titanium powder.

**Figure 3 materials-14-05492-f003:**
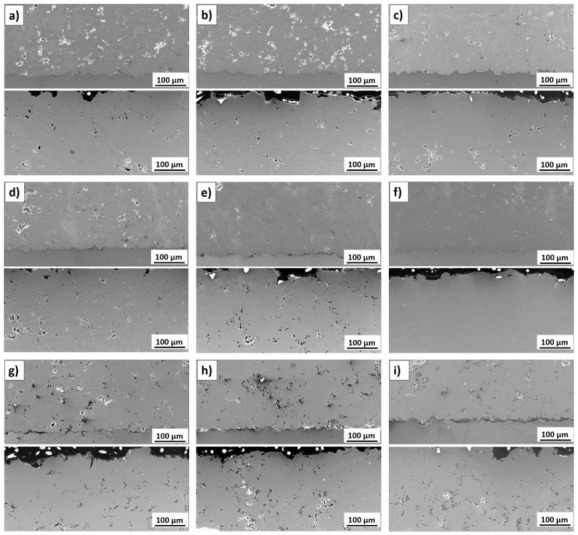
Microstructure of the cold-sprayed Ti coatings (near substrate—upper picture, top—picture below) deposited at standoff distances of: (**a**) 20 mm, (**b**) 30 mm, (**c**) 40 mm, (**d**) 50 mm, (**e**) 60 mm, (**f**) 70 mm, (**g**) 80 mm, (**h**) 90 mm, and (**i**) 100 mm.

**Figure 4 materials-14-05492-f004:**
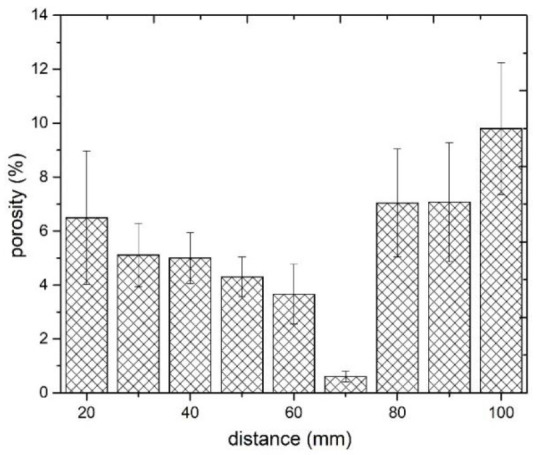
The porosity of the cold-sprayed titanium coatings versus the standoff distance (20–100 mm).

**Figure 5 materials-14-05492-f005:**
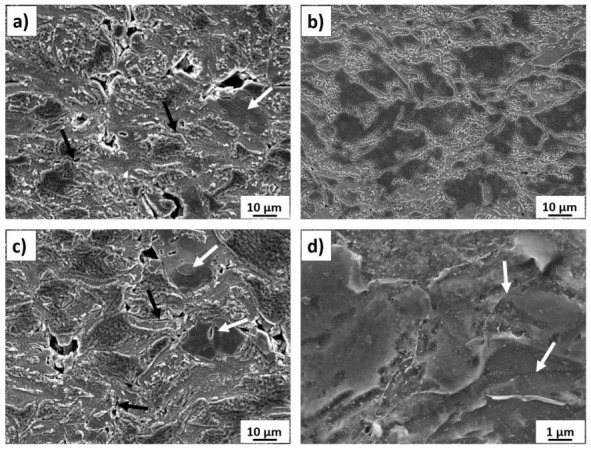
Etched microstructure of the cold-sprayed Ti coatings deposited at standoff distances of: (**a**) 20 mm (white arrows—poorly deformed grains, black arrows—partial bonding among particles), (**b**) 70 mm, (**c**) 100 mm (white arrows—poorly deformed grains, black arrows—partial bonding among particles), (**d**) 70 mm at high-magnification (white arrows—interparticle bonding).

**Figure 6 materials-14-05492-f006:**
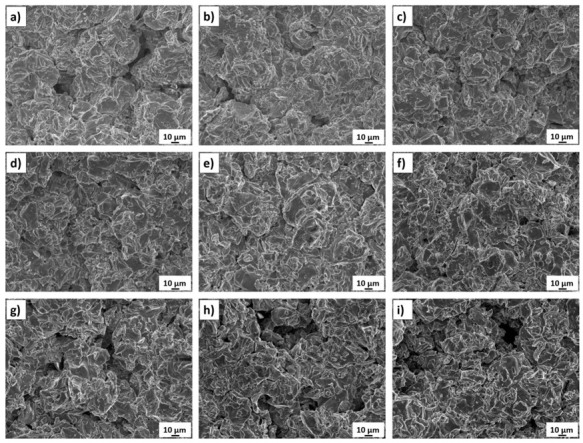
Surface morphologies of the cold-sprayed Ti coatings deposited at standoff distances of: (**a**) 20 mm, (**b**) 30 mm, (**c**) 40 mm, (**d**) 50 mm, (**e**) 60 mm, (**f**) 70 mm, (**g**) 80 mm, (**h**) 90 mm, and (**i**) 100 mm.

**Figure 7 materials-14-05492-f007:**
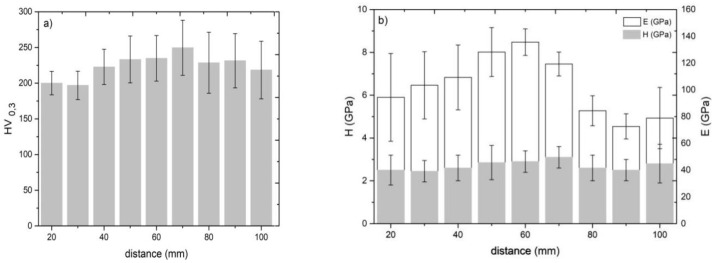
Effect of the standoff distance on: (**a**) microhardness, (**b**) nanohardness, and elastic modulus.

**Figure 8 materials-14-05492-f008:**
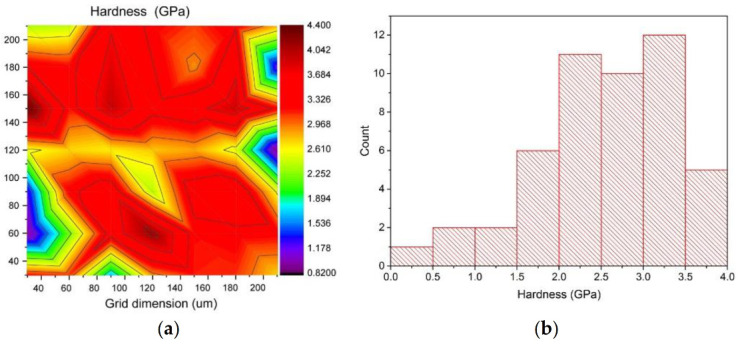
Distribution of the mechanical properties for a Ti coating deposited at a standoff distance of 20 mm: (**a**) hardness map, (**b**) hardness histogram, (**c**) elastic modulus map, (**d**) elastic modulus histogram.

**Figure 9 materials-14-05492-f009:**
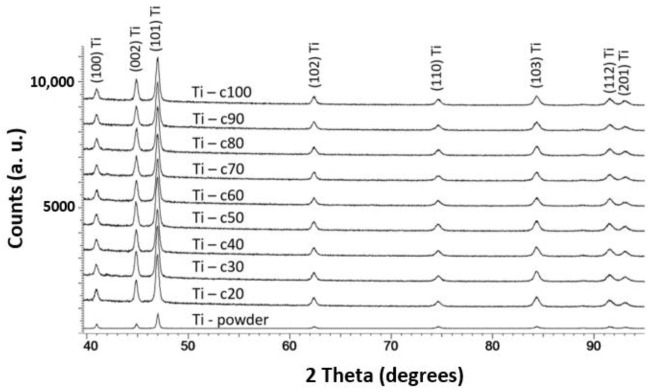
X-ray diffraction patterns of the titanium powder and cold-sprayed titanium coatings deposited at different standoff distances (20–100 mm).

**Figure 10 materials-14-05492-f010:**
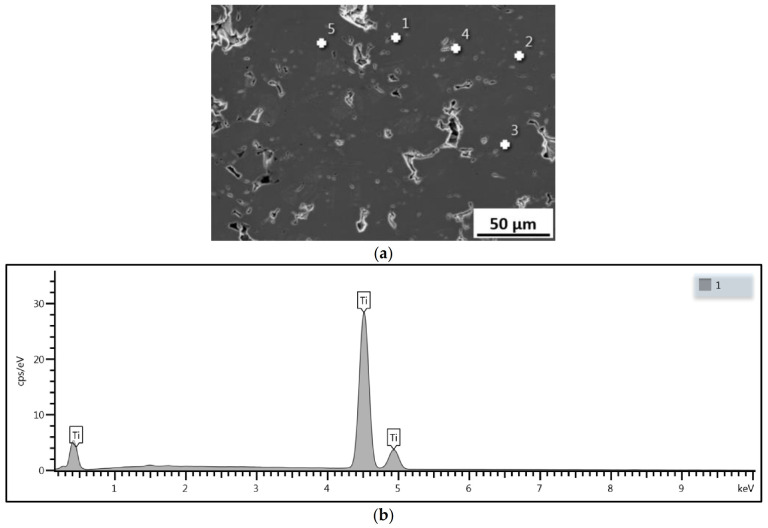
(**a**) Cross-section of a Ti coating sprayed at 100 mm with marked points (1–5) of the EDX analysis (white crosses), (**b**) EDX analysis for point 1.

**Figure 11 materials-14-05492-f011:**
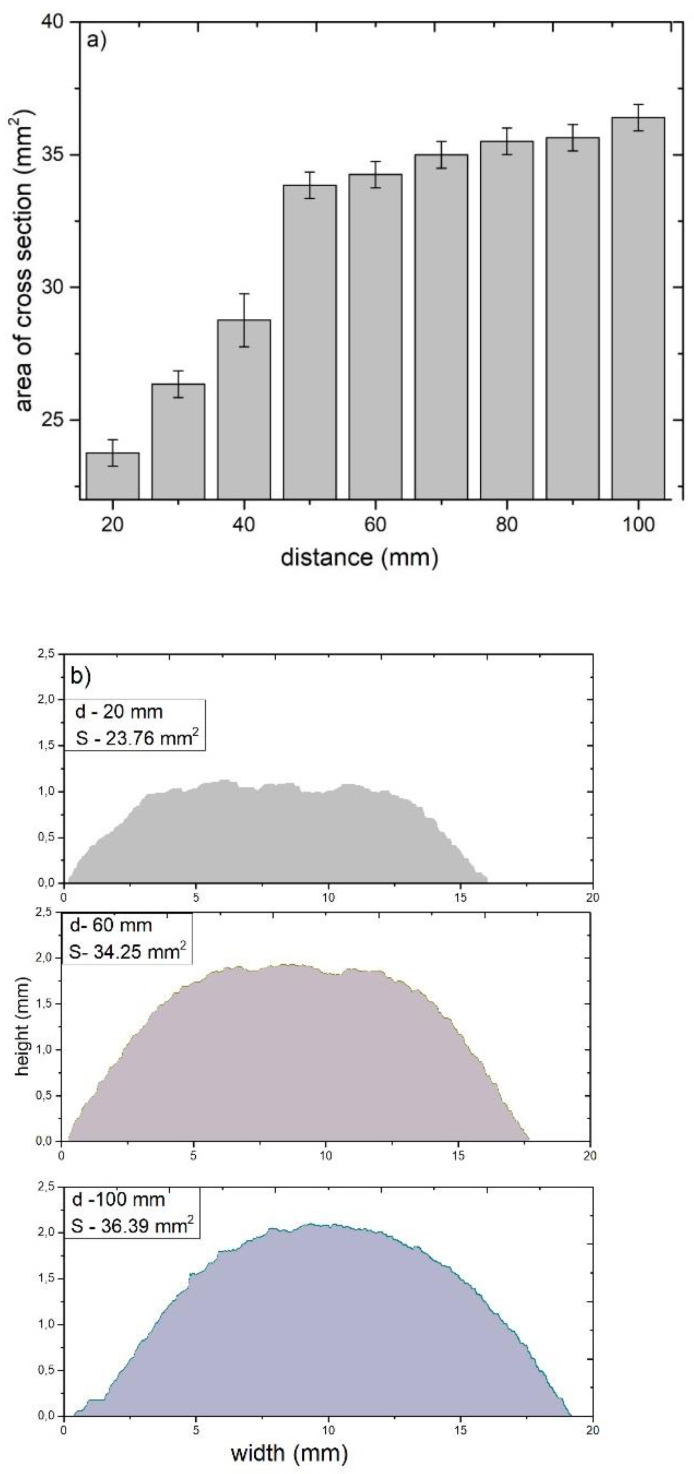
(**a**) Cross-sectional area versus the standoff distance (20–100 mm), (**b**) coating profiles for standoff distances of 20 mm, 60 mm and 100 mm.

**Figure 12 materials-14-05492-f012:**
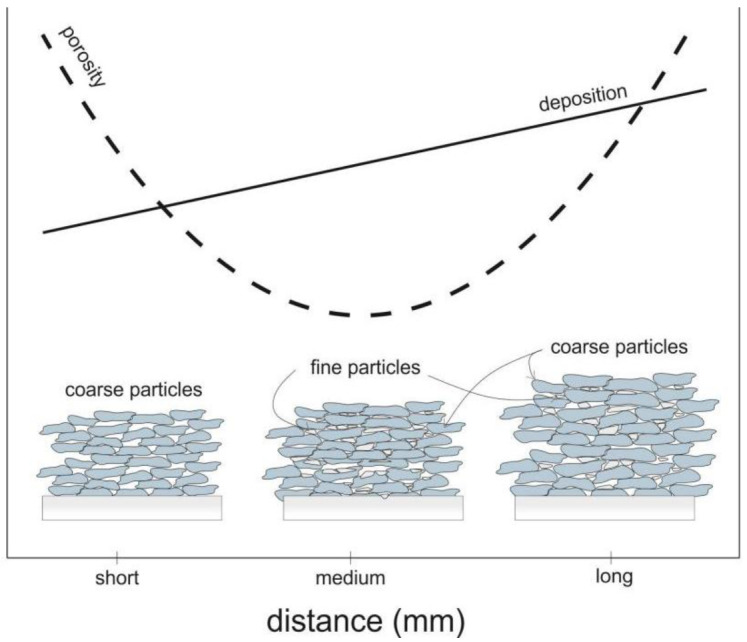
Effect of the increasing standoff distance on the microstructure of the cold-sprayed coating.

**Figure 13 materials-14-05492-f013:**
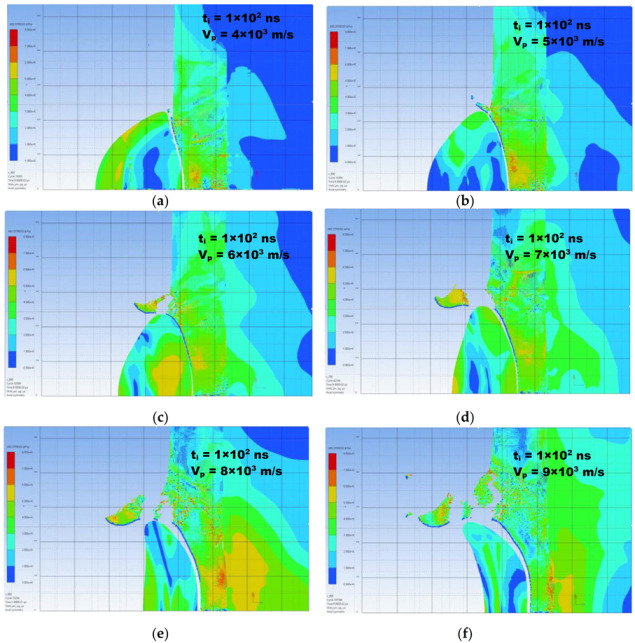
(**a**) Calculated cross-sections of a titanium particle after 100 [ns] from its impact onto a 7075 aluminium alloy plate as a function of the incident particle velocity: (**a**) 400 m/s, (**b**) 500 m/s, (**c**) 600 m/s, (**d**) 700 m/s, (**e**) 800 m/s, and (**f**) 900 m/s.

**Figure 14 materials-14-05492-f014:**
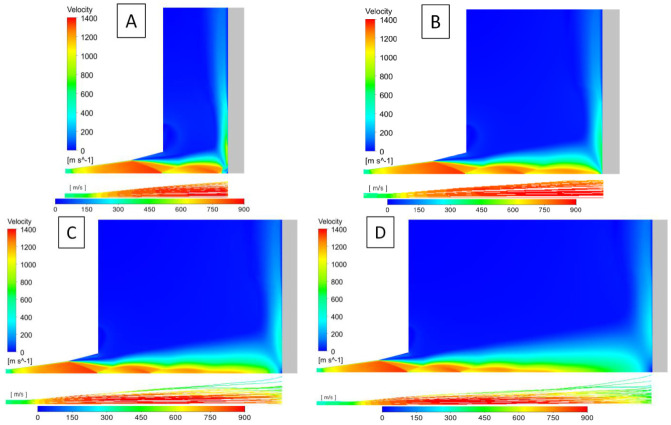
The velocity profiles of the gas stream (left side) and deposited particles (right side) as a function of standoff distance: (**A**)—20 mm, (**B**)—40 mm, (**C**)—60 mm, (**D**)—80 mm.

**Table 1 materials-14-05492-t001:** Surface topography parameters according to ISO 25178.

Coating				Height Parameters			
	S_a_ μm	S_q_ μm	S_sk_	S_ku_	S_p_ μm	S_v_ μm	S_z_ μm
Ti-20	16.98	21.07	0.10	2.67	117.93	73.34	191.30
Ti-30	23.40	29.28	0.11	2.88	98.59	106.30	204.89
Ti-40	20.82	26.22	0.17	2.94	85.04	89.85	174.90
Ti-50	17.89	22.38	0.13	2.99	97.41	76.50	174.41
Ti-60	19.28	23.70	−0.01	2.54	71.54	81.65	153.19
Ti-70	19.13	24.10	−0.31	3.13	73.05	104.74	177.79
Ti-80	16.20	20.57	0.34	3.24	78.77	83.09	161.86
Ti-90	21.07	26.60	−0.02	3.01	84.05	92.78	176.83
Ti-100	18.46	23.03	0.01	2.82	79.24	97.70	176.95

Key: S_a_—arithmetic mean height, S_q_—root mean squared height, S_sk_—skewness, S_ku_—kurtosis, S_p_—maximum peak height, S_v_—maximum valley depth, S_z_—maximum height.

## Data Availability

Data sharing is not applicable.

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
