# Peer review of "Experimental and Numerical Investigations of Titanium Deposition for Cold Spray Additive Manufacturing as a Function of Standoff Distance"

_materials, 2021, doi:10.3390/ma14195492_

Round 1

Reviewer 1 Report

The manuscript has interesting results that would be beneficial for the cold spray community. However, I would like to suggest some amendments as follows:

  1. Since the article is mainly focused on the “influence of standoff distance”, it should be mentioned in the article title.
  2. Line 32 : “…..then increased significantly to 9.8%, which increased hardness of the coatings by 30%”. What increased hardness by 30% ? (sentence is not clear to me)
  3. Authors have used two terms “spray distance” and “standoff distance” throughout the paper, which is confusing (I guess both means the same?). For consistency, I would suggest to use one terminology i.e. “standoff distance” which is more commonly used.
  4. Line 44 to 58 introduces the CS process, which can be reduced, as this process is quite well-know now a days.
  5. What is “bow shock”? few words on this would help reader or a reference.
  6. Motivation behind this work is not very clear from the introduction, author can improve the literature review part (line 68-87), stating the state-of-the-art, gap in the literature, and the importance of this study. In the state-of-the-art, author should include all the work focused on the effect of standoff distance on CS deposits properties.
  7. Section 2. Experimental details can be well structured by making sub sections or at least different paragraphs explaining different experiments methods, e.g. CS process parameters and specimen preparation (including feedstock powder details), phase analysis, microstructural characterization, mechanical properties, deposition efficiency, numerical analysis, etc.
  8. Line 106- Why Al 7075 was chosen as a substrate?
  9. Section 3.1 characteristics of feedstock powder can be moved section 2 experimental methods.
  10. Figure 2: (i) particle size unit should be µm (not um), (ii) in the caption, “Particle size distribution” is preferred over “grain size distribution”. ‘Grain’ may be misleading, which is also mentioned in line 267.
  11. Figure 3: there should be gap between “near substrate” and “top” micrographs.
  12. Figure 5: what is pointed by the white cursor should be mentioned in the figure or in the caption.
  13. Section 3.2 (microstructure) and 3.6 (deposition efficiency) is a bit lengthy, which can written in a concise way and message can more clear for better readability.
  14. Line 476-478, statement regarding thermal residual stress is not completely true. Contribution of quenching stresses can be significant for CS coatings deposited at high temperature and pressure, and stress profile can be very similar to thermal spray processes. For example, see this paper 'Evaluation of residual stresses induced by cold spraying of Ti-6Al-4V on Ti-6Al-4V substrate'.
  15. Table 1: Ti-50mm is missing, but it is being discussed in the text.
  16. Figure 14 and 15 can be merged into one figure

Reviewer 2 Report

The submitted manuscript deals with the numerical/experimental analysis of the effects of the standoff distance in the cold spray deposition of Titanium powder on substrates in aluminum alloy AA7075. The authors studied the powder feedstock granulometry and morphology, microstructure, microhardness, and features of the deposited coatings, and effectiveness of the coating. Finally presented the numerical simulations of the cold spray process were performed.

The manuscript is well-written, and the numerous results reported are satisfactory commented on.

Some remarks are following reported:

  • In the referee’s opinion, the authors should include a short description of the state-of-the-art about numerical modeling of the cold spray process in the Introduction section.
  • In the first paragraph of the introduction, the authors should specify that they are dealing with the additive manufacturing of metals, to avoid possible confusion with the additive manufacturing of polymer, polymer composites, or other materials.
  • Cold spray deposition has been effectively adopted to coat/functionalize different kinds of surfaces. For instance, it is worth reporting on the adoption of this technology to coat polymer or polymer composites, as described in “Metallisation of polymers and polymer matrix composites by cold spray: state of the art and research perspectives”, by Parmar et al., International Materials Review (2021), or in “Metallization of fiber reinforced composite by surface functionalization and cold spray deposition”, Rubino et al., Procedia Manufacturing (2020).
  • The authors should describe the numerical models implemented: load, boundary conditions, impact/contact model, discretization, …
  • The regression equation presented in Figure 11 and on lines 547-548 does not satisfactorily describe the dependence between cross-sectional area and standoff distance. In the referee’s opinion, the equation and the regression curve should be removed.
  • Figures 14 and 15 should be revised including details promoting their readability such as spatial scales and frames of reference, and their dimensions should be homogenized. Moreover, the graphical impact can be improved: the streamlines illustrated on the right side do not promote the comprehension of the simulated data. The authors could merge the left and right images overlapping streamlines and contours.

Reviewer 3 Report

This paper should be of great interest to our readership. However, the authors could significantly improve the manuscript by including additional details about the numerical simulations. In my opinion, at least 4-5 paragraphs must be added in the section of “numerical simulations”. The details of computational domain, geometry (2D or 3D?), boundary and initial conditions, mesh size, numerical schemes (e.g. 2nd order upwind), modeling of wall effects, employed drag and Nu coefficients (have you considered the effects of other forces such as thermophoretic?), particle size distribution modeling (have you used the Rosin-Rammler distribution?), particle injection etc. must be explained. Similarly, the assumptions used for the SPH and ALE simulations must be given.

In addition, the numerical results should be presented in a more scientific way. For example, like the work of Jadidi et al. (M Jadidi, M Mousavi, S Moghtadernejad, A Dolatabadi (2015) A three-dimensional analysis of the suspension plasma spray impinging on a flat substrate, Journal of Thermal Spray Technology 24 (1-2), 11-23), Stokes number should be used to show the particle inflight behavior near the substrate. In this case, the effects of particle velocity, size etc. can be shown easily. In my opinion, the thermal spray community should focus on developing effective nondimensional numbers to explain the physics and predict different parameters, instead of running several simulations for different particles. Plotting the normalized impact velocity (from numerical simulations) as a function of particle diameter for different standoff distances is also very useful. Like the work of Garmeh et al. (S Garmeh, M Jadidi, A Dolatabadi (2020) Three-dimensional modeling of cold spray for additive manufacturing, Journal of Thermal Spray Technology 29 (1), 38-50), the normal impact velocity can be normalized by the critical velocity (which is a function of particle size, temperature, etc. (there are empirical correlation for that)).

Author Response

Dear Reviewer 3,

Thank you for your comments regarding our manuscript entitled " Experimental and Numerical Investigations of Titanium Deposition for Cold Spray Additive Manufacturing". The paper has been improved, taking into consideration your suggestions. We have attached the revised paper with corrections introduced in the text. The added and corrected text is red-marked. Below, please, find enclosed answers to your comments:

This paper should be of great interest to our readership. However, the authors could significantly improve the manuscript by including additional details about the numerical simulations. In my opinion, at least 4-5 paragraphs must be added in the section of “numerical simulations”. The details of computational domain, geometry (2D or 3D?), boundary and initial conditions, mesh size, numerical schemes (e.g. 2nd order upwind), modeling of wall effects, employed drag and Nu coefficients (have you considered the effects of other forces such as thermophoretic?), particle size distribution modeling (have you used the Rosin-Rammler distribution?), particle injection etc. must be explained. Similarly, the assumptions used for the SPH and ALE simulations must be given.

In addition, the numerical results should be presented in a more scientific way. For example, like the work of Jadidi et al. (M Jadidi, M Mousavi, S Moghtadernejad, A Dolatabadi (2015) A three-dimensional analysis of the suspension plasma spray impinging on a flat substrate, Journal of Thermal Spray Technology 24 (1-2), 11-23), Stokes number should be used to show the particle inflight behavior near the substrate. In this case, the effects of particle velocity, size etc. can be shown easily. In my opinion, the thermal spray community should focus on developing effective nondimensional numbers to explain the physics and predict different parameters, instead of running several simulations for different particles. Plotting the normalized impact velocity (from numerical simulations) as a function of particle diameter for different standoff distances is also very useful. Like the work of Garmeh et al. (S Garmeh, M Jadidi, A Dolatabadi (2020) Three-dimensional modeling of cold spray for additive manufacturing, Journal of Thermal Spray Technology 29 (1), 38-50), the normal impact velocity can be normalized by the critical velocity (which is a function of particle size, temperature, etc. (there are empirical correlation for that)).

The required details regarding the numerical model, its boundary conditions and constraints have been added to the publication's text. Unfortunately, we cannot perform any additional simulations due to the completion and settlement of the project founded from EU public sources.

Round 2

Reviewer 3 Report

It seems that the authors have not spent enough time addressing the comments. The authors mentioned that "Unfortunately, we cannot perform any additional simulations due to the completion and settlement of the project founded from EU public sources.". However, I didn't ask for new simulations! They were asked to give more information about their simulations and present their results in a better way (they only need to perform postprocessing using MATLAB, excel....)

Many simulations have been performed but we don't learn anything important from them and we cannot repeat the simulations without knowing, for example, the wall-function or drag and Nu correlations. The authors must know that 1. they must write in such a way that their numerical simulations can be repeated easily by other researchers 2. the reader must learn something from the simulations. Just confirming the experimental results is not enough!

Therefore, the following comments must be addressed:

  1. The details of computational domain (its size), geometry (2D or 3D?), boundary and initial conditions of discrete phase, mesh size, time step for calculation of discrete phase, modeling of wall effects (wall functions), employed drag and Nu coefficients (for example, if spherical drag was used, the results would not be accurate), particle size distribution modeling (have you used the Rosin-Rammler distribution?), etc.
  2. Have you considered the effects of other forces such as thermophoretic?
  3. Stokes number should be used to show the particle inflight behavior near the substrate. 
  4. Plotting the normalized impact velocity (from numerical simulations) as a function of particle diameter for different standoff distances is also very useful. The normal impact velocity can be normalized by the critical velocity (which is a function of particle size, temperature, etc. (there are empirical correlation for that)).

Round 3

Reviewer 3 Report

No comments